# A Split-Client Approach to Second-Order Optimization

## Abstract

Second-order methods promise faster convergence but are rarely used in practice because Hessian computations and decompositions are significantly more expensive than gradient computations. We propose a *split-client* framework where gradients and curvature are computed asynchronously by separate clients. This abstraction captures realistic delays and inexact Hessian updates while avoiding the manual tuning required by Lazy Hessian methods. Focusing on cubic regularization, we show that our approach retains strong convergence guarantees and achieves a provable wall-clock speedup of order $\sqrt{\tau}$, where $\tau$ is the relative time needed to compute and decompose the Hessian compared to a gradient step. Since $\tau$ can be orders of magnitude larger than one in high-dimensional problems, this improvement is practically significant. Empirical results on convex and nonconvex problems confirm that the asynchronous method achieves faster wall-clock convergence while naturally satisfying the bounded-delay and inexactness assumptions.

## 1 Introduction

Second-order optimization methods, which leverage curvature information through the Hessian, are among the most powerful tools in continuous optimization. Classical Newton's method, for example, enjoys local quadratic convergence and strong theoretical guarantees. However, their widespread use in machine learning and large-scale optimization has been limited by a practical barrier: computing and decomposing the Hessian is far more expensive than evaluating gradients. In high-dimensional problems, decomposition often dominates runtime, making second-order methods prohibitively costly despite their superior convergence properties (Conn et al., 2000; Griewank, 1981; Nesterov & Polyak, 2006; Cartis et al., 2011a; 2012b; Curtis et al., 2017; Byrd & Schnabel, 1987; Eisenstat & Walker, 1996; Cartis et al., 2011b; 2012a; 2019).

A natural question arises: *can we retain the benefits of second-order methods while mitigating their computational bottlenecks?*

Recent work has attempted to address this. The *Lazy Hessian* method of Doikov et al. (2023) reduces per-iteration cost by reusing previously computed curvature information, yielding provable arithmetic savings. Other approaches rely on quasi-Newton approximations or auxiliary information (Chayti & Karimireddy, 2022; Chayti et al., 2023) to accelerate convergence. While effective in some regimes, these methods either require manual tuning (e.g., update frequency in Lazy Hessian) or fail to account for the true *wall-clock cost* of Hessian computation and decomposition. As a result, there remains a significant gap between theoretical complexity improvements and actual runtime performance in practice.

In this paper, we propose a new abstraction—the **split-client framework**—that directly targets this gap. In our model, gradients and curvature are provided by independent clients operating in parallel. Gradients are available immediately at the current iterate, while curvature information (Hessian or its approximation) arrives asynchronously, possibly delayed or inexact. This captures the real computational pipeline: gradient updates may run efficiently on accelerators such as GPUs, while curvature and decomposition can be handled separately on CPUs or distributed workers, overlapping with gradient work instead of blocking it.

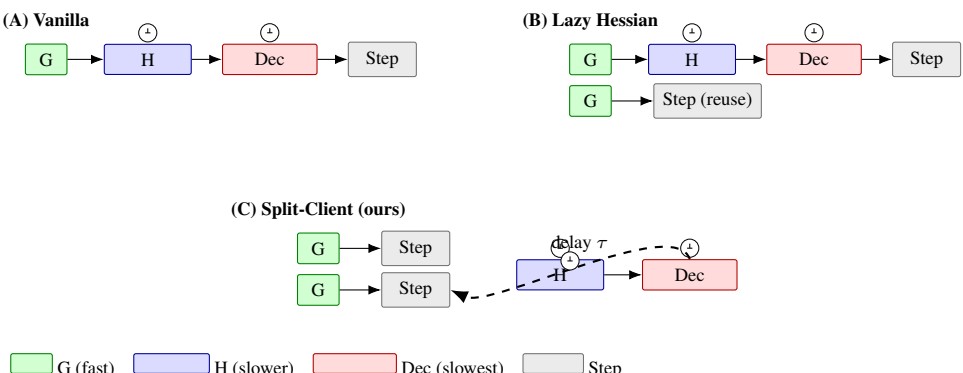

Figure 1: Compact comparison with *relative compute time encoded by box width*. (A) *Vanilla* re-computes and decomposes the Hessian each iteration (H slower than G; Dec slowest, causing block-ing). (B) *Lazy* reuses decompositions between updates (still blocks on refresh and requires tuning). (C) *Split-Client (ours)* deliberately separates fast gradients from slower asynchronous curvature; a dashed arrow labeled "delay $\tau$" and extra spacing make the arrival lag explicit, yielding a $\sqrt{\tau}$ wall-clock speedup.

We instantiate this framework for cubic regularization, a robust second-order method with strong global guarantees (Nesterov & Polyak, 2006; Cartis et al., 2011a; 2012b). Our analysis shows that the asynchronous split-client approach improves wall-clock complexity by a factor of $\sqrt{\tau}$, where $\tau$ is the relative delay for Hessian computation and decomposition compared to a gradient step. Since $\tau$ can be orders of magnitude larger than one in large-scale problems, this speedup translates to significant practical gains. Crucially, unlike Lazy Hessian methods, our approach requires *no manual tuning*, as delays naturally reflect actual compute times.

Our **contributions** are threefold:

1. **A new asynchronous model for second-order optimization.** The split-client abstraction ex-plicitly incorporates delay and inexactness in curvature computation, unifying standard, delayed, and approximate Hessian regimes.

2. **Theoretical guarantees with wall-clock complexity bounds.** We extend cubic regularization analysis to the asynchronous setting, proving convergence rates that highlight the $\sqrt{\tau}$ improve-ment.

3. **Empirical validation.** Experiments on synthetic and real datasets demonstrate that asynchronous curvature consistently outperforms vanilla and Lazy Hessian baselines, both with exact and ap-proximate curvature. A time-profile analysis further confirms that decomposition is the dominant cost our method successfully overlaps.

Overall, this work identifies asynchronous curvature handling as a powerful design principle for modern second-order methods. By decoupling gradient and curvature computation, our framework bridges the gap between theoretical second-order efficiency and practical runtime performance, and opens new avenues for distributed and large-scale optimization.

## 2 PROBLEM SETUP

We consider the unconstrained optimization problem

$$\min_{x \in \mathbb{R}^d} f(x),$$

where $f : \mathbb{R}^d \to \mathbb{R}$ is twice continuously differentiable and possibly non-convex. At iteration $t$, two computational agents operate in parallel:

- The **gradient client** computes the exact gradient at the current iterate:

$$g_t = \nabla f(x_t).$$

- The **curvature client** computes an *approximation* to the Hessian matrix, but may return it with a delay. We denote by $\tau_t \in \mathbb{N}$ the *delay* at iteration $t$, so that the curvature information corresponds to a past iterate $x_{t-\tau_t}$. The returned matrix is

$$\tilde{H}_t = H_{t-\tau_t} + E_t,$$

where $H_{t-\tau_t} = \nabla^2 f(x_{t-\tau_t})$ is the exact Hessian at the delayed iterate, and $E_t$ models the inexactness (e.g., quasi-Newton approximation, subsampling error, or other curvature approximation). **Note** that we don't need to know $E_t$ in practice.

**Initialization with a predefined Hessian.** Before the first asynchronous curvature update arrives, we initialize with a predefined matrix $H_0$ (e.g., a problem-specific surrogate or a previous run's curvature) and use it in the cubic step at $t = 0$. We explicitly model its inexactness:

$$\|H_0 - \nabla^2 f(x_0)\| \le \delta_0.$$

The curvature client exposes $\tilde{H}_0 := H_0$ until a fresher Hessian approximation becomes available.

**Delay as compute & decomposition time.** The delay $\tau_t$ explicitly models that computing the Hessian is significantly more costly than computing the gradient, both in terms of *evaluation time* and the additional *decomposition or factorization* (e.g., spectral or Cholesky) needed to solve the cubic subproblem. This decomposition cost can be substantial, especially in high dimensions, and may dominate the Hessian evaluation time. Our framework incorporates both contributions to the delay, making the performance model more realistic.

**Comparison with Lazy Hessian.** A key distinction with the *Lazy Hessian* method of Doikov et al. (2023) is that, while that approach also reuses a decomposition of the Hessian across multiple steps, it does not include the time needed to *perform* the decomposition in its cost model. In practice, this time can be substantial—especially for high-dimensional problems—since decompositions are often more expensive than a single gradient evaluation. In contrast, our asynchronous framework explicitly accounts for this cost via the delay parameter $\tau_t$.

Moreover, Lazy Hessian requires the practitioner to *tune* the update frequency $m$ (number of iterations between Hessian recomputations). The optimal choice of $m$ depends on the *effective dimension*, defined as the ratio

$$m_{\mathrm{opt}} = \frac{HessCost}{GradCost},$$

which is usually problem-dependent and *unknown* in practice. Our asynchronous approach avoids this tuning entirely: the delay $\tau_t$ naturally emerges from the actual time taken to compute and decompose the Hessian, without requiring manual calibration.

**Asynchronous operation.** The gradient and curvature clients operate independently:

- The delay $\tau_t$ may vary with $t$, reflecting irregular arrival times of Hessian and decomposition updates.

- The same delayed $\tilde{H}_t$ may be reused across several steps until a new update arrives.

Given $g_t$ and the most recent Hessian approximation $\tilde{H}_t$, the cubic step is

$$s_t = \arg\min_{s \in \mathbb{R}^d} \left\{ g_t^\top s + \frac{1}{2} s^\top \tilde{H}_t s + \frac{\rho_t}{6} \|s\|^3 \right\},$$

with $\rho_t > 0$ the cubic regularization parameter, and we update

$$x_{t+1} = x_t + s_t.$$

**Algorithm 1** Split-Client Cubic Regularization with Asynchronous Hessians

---

1: **Input:** Initial point $x_0$, cubic parameter $\rho_t > 0$, predefined $H_0$.
2: Initialize $\tilde{H} \leftarrow H_0$
3: **for** $t = 0, 1, 2, \dots$ **do**
4:     Receive gradient $g_t = \nabla f(x_t)$ from gradient client
5:     **if** curvature client returns a new Hessian approximation **then**
6:         Receive $\tilde{H} \leftarrow H_{t-\tau_t} + E_t$
7:     Compute cubic step:

$$s_t = \arg\min_s \left\{ g_t^\top s + \frac{1}{2} s^\top \tilde{H} s + \frac{\rho_t}{6} \|s\|^3 \right\}$$

8:     Update: $x_{t+1} = x_t + s_t$

---

Special cases:

- $\tau_t = 0$, $\delta_0 = \delta = 0$ recovers the standard cubic regularized Newton method.
- $\tau_t > 0$, $\delta = 0$ yields delayed-Hessian cubic regularization.
- $\tau_t \geq 0$, $\delta > 0$ models quasi-Newton or other approximate-Hessian cubic methods, including possible initialization from a surrogate Hessian.

This formulation explicitly accounts for three sources of deviation from the exact Newton step: (i) the *initial inexactness* $\delta_0$ from $H_0$, (ii) the *delay* $\tau_t$ (Hessian computation and decomposition time), and (iii) the *inexactness* $E_t$ of subsequent curvature updates.

**Remark.** Although we instantiate our framework with cubic regularization (Nesterov & Polyak, 2006; Cartis et al., 2011a), the split-client abstraction applies broadly to other second-order algorithms such as trust-region methods (Conn et al., 2000), Levenberg–Marquardt schemes (Levenberg, 1944; Marquardt, 1963), or higher-order regularization techniques (Agarwal et al., 2017). We chose cubic Newton primarily for its well-understood global convergence guarantees and for clarity of exposition.

## 3 RELATED WORK

**Second-order methods.** Second-order methods exploit curvature information to accelerate convergence beyond what is possible with first-order methods. Classical Newton and trust-region methods (Conn et al., 2000) achieve quadratic local convergence but require solving large linear systems at each step. Refinements such as self-concordant analysis (Nesterov & Nemirovski, 1994; Bach, 2010; Sun & Tran-Dinh, 2019; Dvurechensky & Nesterov, 2018) and Hessian stability bounds (Karimireddy et al., 2018) provide stronger complexity guarantees. Among globalization strategies, cubic regularization (Nesterov & Polyak, 2006; Cartis et al., 2011a) has emerged as particularly robust, with extensions including contracting-point steps (Doikov & Nesterov, 2020; 2022) and gradient regularization (Mishchenko, 2021; Doikov & Nesterov, 2021). Our work builds on this line by adapting cubic regularization to the asynchronous setting.

**Lazy Hessian updates.** The *Lazy Hessian* method of Doikov et al. (2023) reduces arithmetic complexity by reusing Hessian decompositions across multiple steps. While effective in theory, this approach requires tuning the update frequency based on the unknown "effective dimension" (ratio of Hessian to gradient cost) and does not account for the often dominant cost of Hessian *decomposition*. Our framework addresses both issues: the delay parameter $\tau$ naturally reflects actual Hessian and decomposition times, eliminating the need for manual tuning and providing realistic wall-clock predictions.

**Quasi-Newton and inexact Hessians.** Quasi-Newton methods (Broyden, 1967; Fletcher, 1970; Shanno, 1970; Nocedal & Wright, 1999) approximate the Hessian via low-rank updates such as BFGS or SR1, achieving cheap iterations with superlinear local rates. Recent work has extended

cubic regularization to quasi-Newton updates: Kamzolov et al. (2023) analyze cubic regularized quasi-Newton methods, showing that these can be interpreted as *inexact Hessians* with adaptive error control. This interpretation is directly compatible with our model, in which the curvature client may deliver a $\delta$-inexact Hessian.

**Auxiliary information and helper frameworks.** Several recent frameworks explore leveraging additional information sources. Chayti & Karimireddy (2022) study optimization with access to auxiliary gradients, while Chayti et al. (2023) develop a unified convergence theory for stochastic and variance-reduced cubic Newton methods, incorporating both lazy and inexact Hessians. These approaches share our multi-oracle spirit, but unlike ours they do not explicitly account for the *time cost* of curvature computation and decomposition.

**Asynchrony and delays.** Asynchrony has been extensively studied in first-order distributed optimization, e.g., Hogwild! (Niu et al., 2011) and its extensions (Lian et al., 2015), but has received little attention in the second-order setting. Our work brings this perspective to cubic regularization by explicitly modeling delayed and inexact curvature, analyzing its effect on both convergence and wall-clock complexity.

**Summary.** In contrast to these prior directions, our proposed *split-client* framework unifies and extends them by:

1. explicitly modeling delays that reflect the true computational cost of Hessian evaluation and decomposition,

2. allowing inexact curvature to subsume quasi-Newton and approximate Hessian methods, and

3. analyzing cubic regularization under this asynchronous abstraction, with provable convergence and wall-clock complexity guarantees.

Our theoretical and empirical findings demonstrate that asynchronous curvature consistently improves wall-clock performance compared to both vanilla cubic Newton and Lazy Hessian baselines.

## 4 THEORETICAL ANALYSIS

We analyze the convergence and wall-clock complexity of Algorithm 1, extending cubic regularization to handle *delayed* and *inexact* second-order information, including inexact initialization.

### 4.1 ASSUMPTIONS

(A1) **Smoothness:** $f$ is twice continuously differentiable and has $L$-Lipschitz continuous Hessian:
$$\|\nabla^2 f(x) - \nabla^2 f(y)\| \leq L\|x - y\|, \quad \forall x, y \in \mathbb{R}^d.$$

(A2) **Bounded Delays:** The Hessian information at iteration $t$ is from $x_{t-\tau_t}$ with
$$0 \leq \tau_t \leq \tau, \quad \forall t,$$
where $\tau$ models the (finite) time to compute and decompose the Hessian.

(A3) **Inexact Initialization:** The initial Hessian approximation $H_0$ satisfies
$$\|H_0 - \nabla^2 f(x_0)\| \leq \delta_0,$$
and is used until the first curvature update arrives.

(A4) **Inexact Hessians:** Subsequent approximations satisfy
$$\|\tilde{H}_t - \nabla^2 f(x_{t-\tau_t})\| \leq \delta, \quad \forall t,$$
with $\delta \geq 0$ accounting for quasi-Newton or other approximations.

(A5) **Finite Initial Suboptimality:** Let
$$F_0 = f(x_0) - f^\star, \quad f^\star = \inf_x f(x),$$
and assume $F_0 < \infty$.

**Discussion.** These assumptions are standard in the analysis of second-order methods. Smoothness (A1) ensures the Hessian does not change arbitrarily fast and is routinely imposed in cubic regularization analyses (Nesterov & Polyak, 2006; Cartis et al., 2011a; 2012b). Bounded delays (A2) reflect the practical reality that Hessian computation and decomposition require finite time that is typically much greater than that of the gradient, which is a natural assumption in asynchronous or distributed settings (Niu et al., 2011; Lian et al., 2015). Assumptions (A3)–(A4) model inexact curvature: the initialization error $\delta_0$ accounts for using any surrogate Hessian $H_0$ before the first update (e.g., the zero matrix or identity, if no better choice is available), while $\delta$ captures the persistent inexactness from quasi-Newton updates or approximate factorizations (Eisenstat & Walker, 1996; Kamzolov et al., 2023). Finally, finite initial suboptimality (A5) is a mild condition, ensuring that optimization starts from a point with bounded objective gap, which is always satisfied in practice.

## 4.2 OPTIMALITY MEASURE

Following Nesterov & Polyak (2006); Cartis et al. (2011a), define

$$\mu_\rho(x) = \max\left( \|\nabla f(x)\|^{3/2}, \frac{-\lambda_{\min}(\nabla^2 f(x))^3}{\rho^{3/2}} \right),$$

where $\rho > 0$ is the cubic regularization parameter. A small $\mu_\rho(x)$ implies approximate first- and second-order stationarity; in particular,

$$\frac{1}{T} \sum_{t=0}^{T-1} \mu_\rho(x_t) \leq \varepsilon^{3/2}$$

ensures $\|\nabla f(x_t)\| \leq \varepsilon$ and $\lambda_{\min}(\nabla^2 f(x_t)) \geq -\sqrt{\rho\varepsilon}$ on average.

## 4.3 ITERATION COMPLEXITY

With these assumptions in place, we now present the main theoretical result of this work. The following theorem establishes the convergence rate of our split-client cubic regularization method under delayed and inexact curvature information.

---

**Theorem 4.1.** *Under (A1)–(A5), the iterates of Algorithm 1 with $\rho \geq \max\left( L, 20\tau L \right)$ satisfy*

$$\frac{1}{T} \sum_{t=0}^{T-1} \mu_\rho(x_t) = \mathcal{O}\left( \frac{\sqrt{\rho}\, F_0}{T} + \frac{\delta_0^3 \tau_0}{\rho^{3/2} T} + \frac{(T-\tau_0)}{T} \frac{\delta^3}{\rho^{3/2}} \right).$$

*Choosing the optimal $\rho$ yields*

$$\frac{1}{T} \sum_{t=0}^{T-1} \mu_\rho(x_t) = \mathcal{O}\left( \frac{(1+\sqrt{\tau})\sqrt{L}\, F_0 + \tau_0^{1/4}(F_0\delta_0)^{3/4}}{T} + \frac{(T-\tau_0)}{T} \frac{(F_0\delta)^{3/4}}{T^{3/4}} \right).$$

---

**Interpretation.** The $\delta_0$ term models the initial curvature error before the first asynchronous update, scaled by the initial delay $\tau_0 \leq \tau$ until fresh curvature arrives. For large $\tau$ and small $\delta_0$, the main term remains the $\mathcal{O}(\sqrt{\tau L}\, F_0/T)$ dependence from the exact-Hessian delayed setting.

To ensure $\frac{1}{T} \sum_{t=0}^{T-1} \mu_\rho(x_t) \leq \varepsilon^{3/2}$ we need at most

$$T(\varepsilon) = \mathcal{O}\left( \frac{(1+\sqrt{\tau})\sqrt{L}\, F_0 + \tau_0^{1/4}(F_0\delta_0)^{3/4}}{\varepsilon^{3/2}} + \frac{F_0\delta}{\varepsilon^2} \right)$$

iterations.

The proof is deferred to Appendix B.2.

As a sanity check, when there are no delays ($\tau = 0$) and no inexactness ($\delta = \delta_0 = 0$), we recuperate the iteration complexity of Cubic Newton $\mathcal{O}\left( \frac{\sqrt{L}\, F_0}{\varepsilon^{3/2}} \right)$.

## 4.4 WALL-CLOCK COMPLEXITY

We normalize the cost of one gradient (and cubic step given a decomposition of the curvature) to 1. Let $c_H$ be the Hessian computation time and $c_F$ the decomposition time, with $\tau = c_H + c_F$.

For *ease of comparison*, we consider the case where there is no inexactness $\delta = 0$ and large delays $\tau \gg 1$ in this section.

**Wall-clock models.**

- **Asynchronous split-client cubic (ours):** Curvature work overlaps with gradient steps, so $\text{Time}_{\text{async}}(T) = T$.
- **Vanilla cubic Newton:** Curvature work blocks each step, so $\text{Time}_{\text{vanilla}}(T) = T(\tau + 1)$.
- **Lazy Newton (period $p$):** Refresh the curvature in every $p$ step:

$$\text{Time}_{\text{lazy}}(T) = T + \frac{T}{p}\tau.$$

**From iterations to time.** With $\delta = 0$ and $\delta_0 = 0$, Theorem 4.1 gives:

$$T_{\text{async}}(\varepsilon) \asymp \frac{(1 + \sqrt{\tau})\sqrt{L}\,F_0}{\varepsilon^{3/2}}, \quad T_{\text{vanilla}}(\varepsilon) \asymp \frac{\sqrt{L}\,F_0}{\varepsilon^{3/2}}.$$

Thus:

$$\text{Time}_{\text{async}}(\varepsilon) \asymp \frac{(1 + \sqrt{\tau})\sqrt{L}\,F_0}{\varepsilon^{3/2}}, \quad \text{Time}_{\text{vanilla}}(\varepsilon) \asymp (1 + \tau)\frac{\sqrt{L}\,F_0}{\varepsilon^{3/2}}.$$

For large $\tau$, their ratio is $1/\sqrt{\tau}$: our method is faster by a $\sqrt{\tau}$ factor.

**Async vs. Lazy Newton.** If $p$ reflects only Hessian compute time $c_H$ (ignoring decomposition), then $\tau = p + \tau_d$ with $\tau_d \geq 0$, so typically $\tau \geq p$. In this regime,

$$\frac{\text{Time}_{\text{lazy}}(\varepsilon)}{\text{Time}_{\text{async}}(\varepsilon)} = \frac{\sqrt{p} + \dfrac{\tau}{\sqrt{p}}}{1 + \sqrt{\tau}} \geq \frac{2\sqrt{\tau}}{1 + \sqrt{\tau}} \overset{\text{if } \tau \geq 1}{\geq} 1,$$

with strict inequality when $\tau > 1$. Lazy Hessian also requires manually setting $p$; the optimal $p$ depends on the *effective dimension HessCost/GradCost*, which is usually problem-dependent and unknown. Our asynchronous method completely avoids this tuning, as $\tau_t$ reflects the actual compute and decomposition time.

## 5 EXPERIMENTS

We evaluate our split-client cubic framework against *vanilla cubic Newton* (fresh Hessian + decomposition each step) and the *Lazy Hessian* approach (Doikov et al., 2023). We also consider quasi-Newton curvature (L-BFGS). All methods share the same initialization and are tuned over a grid of cubic parameters $\rho$; Lazy additionally tunes the update period $p$.

**Tasks and metrics.** We run logistic regression on (i) synthetic Gaussian data and (ii) the `a1a` dataset (LIBSVM). We report *loss vs. wall-clock time*, which reflects the true cost of gradients, Hessians, decompositions, and cubic solves.

**Cubic Newton (exact curvature).** Figure 2 shows synthetic (left) and `a1a` (right). **Async** converges fastest; **Lazy** helps vs. vanilla but trails async even when $p \approx d$; **Vanilla** is the slowest due to blocking curvature work.

**Quasi-Newton + cubic (approximate curvature).** Figure 3 repeats the comparison with L-BFGS curvature. **Async** remains best, confirming that overlapping curvature updates helps even with inexact Hessians. **Lazy** improves over vanilla but requires $p$ tuning and still lags behind async.

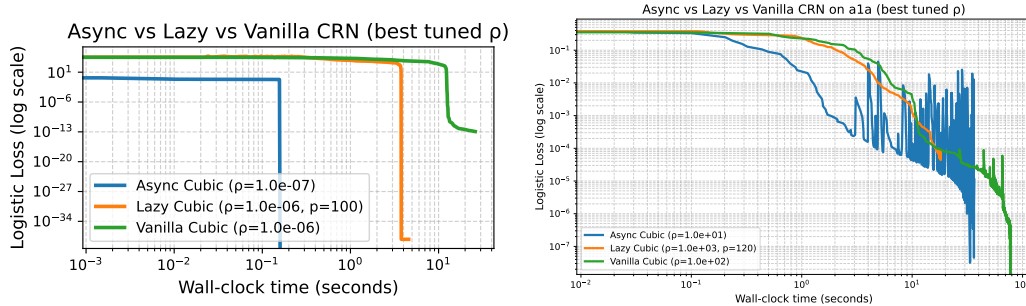

Figure 2: **Cubic Newton (exact curvature):** *Left*—synthetic logistic regression; *Right*—a1a. Async (split-client) converges fastest in wall-clock; Lazy (tuned $p$) helps vs. Vanilla but remains slower than Async.

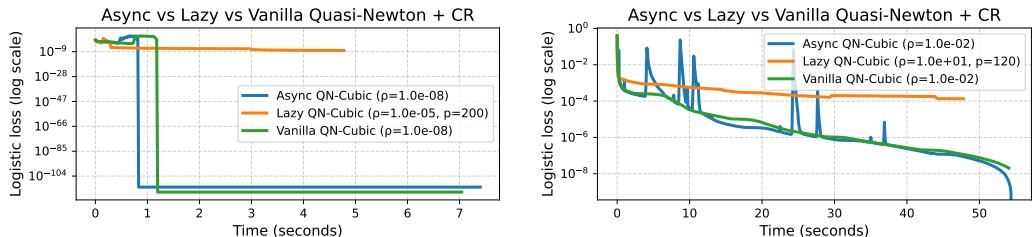

Figure 3: **Quasi-Newton + cubic (L-BFGS curvature):** *Left*—synthetic; *Right*—a1a. Async dominates in wall-clock time; Lazy improves over Vanilla but is sensitive to $p$ and still trails Async.

**Time profiling.** Figure 4 profiles per-iteration costs vs. dimension. *Decomposition* dominates runtime at moderate $d$, while the cubic subproblem (given a decomposition) is comparatively cheap. This validates our delay model: the time to *compute and decompose* curvature must be accounted for. Lazy Hessian does not include decomposition time in its cost model, whereas our asynchronous framework captures it via the delay $\tau$ without any tuning of $p$.

**Takeaway.** Across datasets and curvature models, async cubic methods achieve the fastest wall-clock convergence. Lazy improves over vanilla but depends on the unknown effective dimension (HessianCost/GradientCost) to set $p$, and still underperforms async in time.

## 6 LIMITATIONS AND FUTURE WORK

While our results highlight clear benefits of the asynchronous split-client cubic framework, several limitations and open directions remain:

- **Theoretical assumptions.** Our analysis relies on assumptions that are standard in the study of second-order methods, namely bounded delays and the Lipschitz continuity of the Hessian. These assumptions are reasonable in our setting since the delay naturally corresponds to the extra time required to compute the Hessian compared to the gradient, which is inherently bounded. That said, it would be valuable to relax these assumptions in future work. In particular, instead of assuming a strict uniform bound on the delay, one could aim to develop convergence results that depend on more descriptive statistics of the delay, such as its average or variance. This would better reflect the behavior observed in practice, where delays may vary but remain well-behaved on average. Another interesting direction is to extend the theory beyond globally Lipschitz Hessians to settings with weaker smoothness properties.

- **Initialization.** We model initialization with a $\delta_0$-inexact Hessian $H_0$, but practical strategies such as problem-specific preconditioners, diagonal or block-diagonal surrogates, or

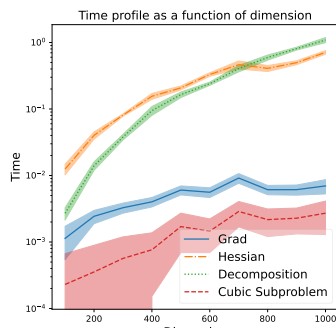

Figure 4: **Time profiling vs. dimension:** Decomposition rapidly becomes the dominant cost, while the cubic subproblem (once decomposed) is relatively cheap. This explains Async's advantage: curvature work is overlapped rather than blocking.

warm-started curvature from previous runs deserve deeper investigation, both empirically and theoretically.

- **Scalability of curvature.** Our method still assumes access to a (possibly approximate) Hessian or a quasi-Newton surrogate. Extending the framework to settings where only stochastic curvature information is available, or where exact decompositions are infeasible, would broaden applicability to very large-scale learning problems.

- **Distributed and hardware aspects.** In our experiments, delays arose naturally from the true computational cost of Hessian evaluation and decomposition, confirming our modeling assumptions. However, we did not yet explore heterogeneous or distributed deployments (e.g., GPUs for gradients, CPUs or separate nodes for curvature). Real-world implementations could reveal additional trade-offs involving communication, synchronization, and resource scheduling, and may also open new opportunities for acceleration.

- **Adaptive strategies.** While our approach avoids the manual tuning required by Lazy Hessian updates, performance is still sensitive to the choice of the cubic parameter $\rho$. Designing adaptive policies for $\rho$, and incorporating techniques such as variance reduction for stochastic gradients, remains an open avenue for improvement.

Future work will address these directions: developing adaptive parameter selection strategies, extending the framework to stochastic and large-scale regimes, and validating wall-clock gains in distributed or heterogeneous learning systems.

## 7 CONCLUSION

We presented a split-client framework for cubic regularization that decouples gradient and curvature computation, enabling Hessian information—possibly delayed and inexact—to be incorporated asynchronously. Our analysis explicitly accounts for delay, initialization error, and curvature inexactness, and establishes a provable $\sqrt{\tau}$ *wall-clock* speedup over classical cubic Newton, while removing the manual tuning required by Lazy Hessian methods.

Empirically, our asynchronous approach consistently outperforms both vanilla and Lazy baselines under exact and approximate curvature. Profiling further shows that Hessian decomposition dominates runtime, and that overlapping curvature computation with gradient updates is the key mechanism driving the observed acceleration.

Beyond cubic regularization, the split-client abstraction extends naturally to a broad family of second-order methods, including trust-region, quasi-Newton, and low-rank or sketched curvature variants, as well as to heterogeneous computing environments where gradient and curvature computations can be performed on different devices. We hope this perspective motivates further exploration of asynchronous second-order optimization, both in theory and in practical large-scale learning systems.

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

# A  ADDITIONAL EXPERIMENTS: SMOOTH NONCONVEX REGRESSION AND EMPIRICAL STATISTICS

We further conducted a detailed study on a smooth nonconvex regression task to assess the empirical behavior of the proposed asynchronous cubic-regularized Newton (Async-CRN) method compared with its Lazy and Vanilla counterparts.

**Experimental setup.**   We consider the smooth nonconvex objective

$$f(x) = \frac{1}{2n}\|Ax - b\|^2 + \lambda \sum_{i=1}^{d} \phi(x_i), \qquad \phi(t) = \frac{t^2}{1 + t^2},$$

where $A \in \mathbb{R}^{n \times d}$ and $b \in \mathbb{R}^n$ are generated synthetically with $n = 5000$, $d \in \{10, 50, 100, 200, 400, 800, 1600\}$, and $\lambda = 10^{-2}$. This objective is nonconvex but smooth, and its curvature structure captures realistic settings where gradients are cheap to evaluate while second-order information (Hessian computation and factorization) is substantially more expensive.

We compare three algorithms:

- **Async Cubic:** our proposed split-client asynchronous method, where curvature is computed on a separate thread that runs concurrently with first-order updates.
- **Lazy Cubic:** a baseline that reuses stale curvature for a fixed number of iterations.
- **Vanilla Cubic:** the standard synchronous cubic-regularized Newton method recomputing curvature at every iteration.

All methods use a cubic regularization parameter $\rho$ tuned on a logarithmic grid, and start from the same initialization.

**Note.**   All methods share the same computational environment. For fairness, wall-clock time includes both gradient and curvature computations. The asynchronous method differs only in that curvature is computed concurrently in a separate thread, allowing overlap without skipping work.

**Wall-clock convergence.**   Figure 5 shows the evolution of the objective value versus wall-clock time. Async-CRN achieves the fastest decrease in loss, converging significantly earlier than both Lazy and Vanilla variants. The Lazy method provides partial improvement but eventually stalls as its curvature becomes outdated, while Vanilla CRN remains slower due to its blocking curvature computations. This confirms that asynchronous curvature overlap translates directly into measurable wall-clock acceleration in regimes where curvature is the computational bottleneck.

**Memory overhead.**   We also tracked the peak memory usage of each method as the problem dimension increases. As shown in Figure 6, the asynchronous method requires slightly higher memory, at most about twice that of the Vanilla version. This overhead arises because Async-CRN stores one additional curvature matrix while computing a new one, which is consistent with the design of the split-client model.

**Delay and inexactness.**   To empirically examine the assumptions used in our analysis, we measured both the curvature delay $\tau_t$ (the number of iterations between successive curvature updates) and the inexactness of the Hessian approximations arising from randomized SVD truncation. Figure 7 shows that the delay remains *uniformly bounded*, fluctuating between 2 and 9 iterations, which directly validates our bounded-delay assumption. Figure 8 reports the Frobenius-norm error $\|\tilde{H}_t - H_t\|_F$, showing that the inexactness is also stably bounded over time and does not accumulate across updates. These empirical observations confirm that the theoretical assumptions of bounded delay and bounded Hessian error hold naturally in practice, without additional enforcement.

**Summary.**   Across all metrics, the asynchronous cubic method consistently achieves the best trade-off between convergence speed, computational efficiency, and memory cost. The boundedness of curvature delay and inexactness observed in practice directly supports the theoretical assumptions of our analysis, while the modest memory overhead confirms that the method remains resource-efficient in realistic computational settings.

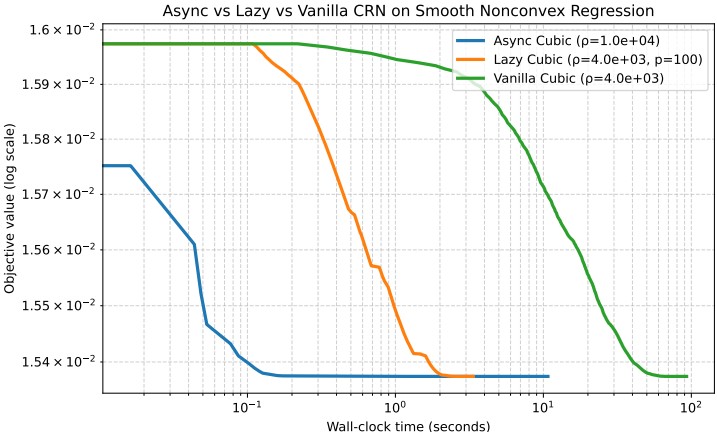

Figure 5: Objective value versus wall-clock time for smooth nonconvex regression. Async cubic converges significantly faster than both Lazy and Vanilla cubic methods.

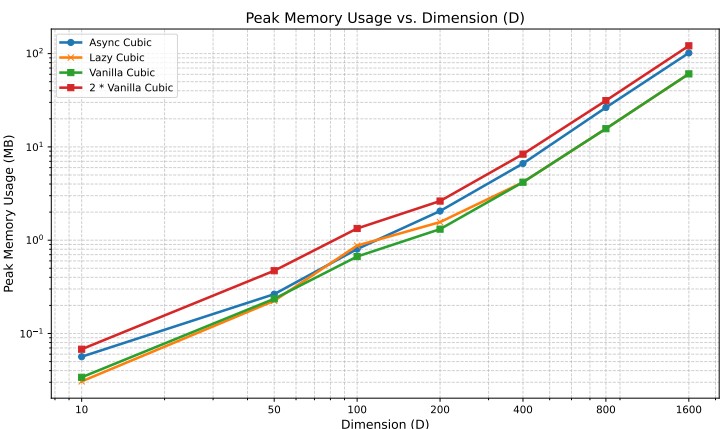

Figure 6: Peak memory usage versus problem dimension. The asynchronous method has modest memory overhead (no more than twice that of the Vanilla method) due to maintaining one additional curvature matrix during asynchronous updates.

**Addition experiment: One-layer** tanh **regression (smooth nonconvex).** We evaluate Async, Lazy, and Vanilla cubic-regularized Newton on a smooth nonconvex regression task defined by the scalar model

$$f(x) \ = \ \frac{1}{2n}\sum_{i=1}^{n}\bigl(\tanh(a_i^\top x) - y_i\bigr)^2,$$

with $x \in \mathbb{R}^d$. We generate a synthetic dataset with $n = 1000$ samples and $d = 500$ features. The design matrix $X \in \mathbb{R}^{n \times d}$ has i.i.d. $\mathcal{N}(0,1)$ entries, then columns are $\ell_2$-normalized. We draw a ground-truth parameter $x_\star \sim \mathcal{N}(0, I_d)$ and set

$$z = Xx_\star, \qquad y = \tanh(z) + \varepsilon, \quad \varepsilon \sim \mathcal{N}(0, 0.05^2 I_n).$$

All methods start from $x_0 = \mathbf{0}$.

**Derivatives and curvature.** The gradient and *exact* Hessian used in the experiments are

$$\nabla f(x) \ = \ \frac{1}{n} X^\top \bigl[(\tanh(Xx) - y) \odot \bigl(1 - \tanh^2(Xx)\bigr)\bigr],$$

$$\nabla^2 f(x) \ = \ \frac{1}{n} X^\top \operatorname{diag}(\alpha)\, X, \ \alpha_i \ = \ \bigl(1 - \tanh^2(z_i)\bigr)^2 - 2\tanh(z_i)\bigl(\tanh(z_i) - y_i\bigr)\bigl(1 - \tanh^2(z_i)\bigr),$$

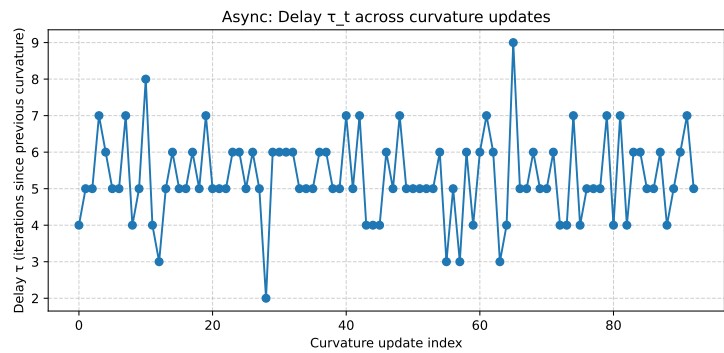

Figure 7: Observed curvature delay $\tau_t$ across updates for the asynchronous method. Delays remain uniformly bounded (between 2 and 9 iterations), validating the bounded-delay assumption used in the analysis.

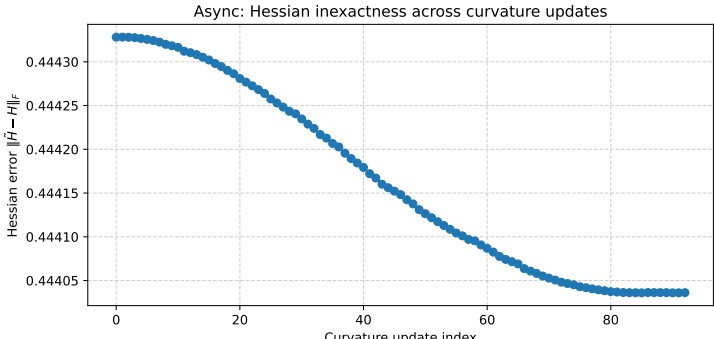

Figure 8: Hessian inexactness $\|\tilde{H}_t - H_t\|_F$ across curvature updates. The error remains stably bounded, confirming the practical validity of the inexactness assumption (A4) for randomized SVD curvature approximations.

with $z = Xx$ and $\odot$ denoting elementwise multiplication. Thus the curvature has the familiar $X^\top \mathrm{diag}(\cdot) X$ structure, but with nonconvex coefficients $\alpha_i$. For all three methods, we solve the cubic step in a low-rank spectral basis of $\nabla^2 f(x)$ obtained by randomized SVD with target rank $r = \min\{50, d\}$.

We tune the cubic parameter $\rho$ on a logarithmic grid $\rho \in \{10^{-2}, 0.1, 1, 10, 10^2, 10^3\}$ and run each method for MAX_ITERS$= 700$ iterations.

**Metric and figure.** We report the objective value $f(x_t)$ versus *wall-clock time* (seconds). The resulting curves are shown in Fig. 9.

The tanh regression task is smooth yet nonconvex, with a Hessian that is costly to form and factorize relative to gradients. In this regime, **asynchronous overlap of curvature computation provides tangible wall-clock gains**: Async converges in substantially less time than Lazy and Vanilla. Lazy benefits from reuse but eventually plateaus as its curvature ages, while Vanilla pays the full blocking cost every iteration. These findings mirror our main results and reinforce the claim that when curvature is the dominant cost, the split-client design translates per-iteration advantages into actual wall-clock speedups.

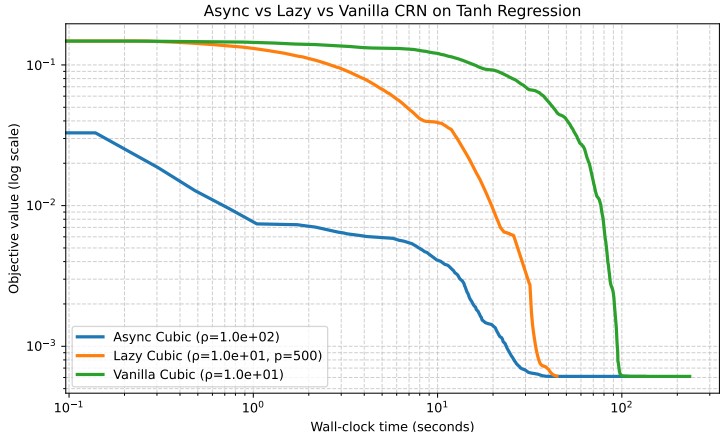

Figure 9: One-layer $\tanh$ regression (exact Hessian): objective value versus wall-clock time. Async achieves the fastest decrease in loss, Lazy improves over Vanilla initially but slows when curvature becomes stale, and Vanilla is consistently slower due to blocking curvature computation each step.

## B  MISSING PROOFS

### B.1  CUBIC NEWTON PRELIMINARIES

We consider the general problem

$$\min_{\boldsymbol{x}\in\mathbb{R}^d} f(\boldsymbol{x})$$

Where $f$ is twice differentiable with $L$-Lipschitz Hessian, i.e.:

$$\|\nabla^2 f(\boldsymbol{x}) - \nabla^2 f(\boldsymbol{y})\| \le L\|\boldsymbol{x} - \boldsymbol{y}\|, \qquad \forall \boldsymbol{x}, \boldsymbol{y} \in \mathbb{R}^d. \tag{1}$$

As a direct consequence of equation 1 (see (Nesterov & Polyak, 2006; Nesterov, 2018)) we have for all $\boldsymbol{x}, \boldsymbol{y} \in \mathbb{R}^d$:

$$\|\nabla f(\boldsymbol{y}) - \nabla f(\boldsymbol{x}) - \nabla^2 f(\boldsymbol{x})(\boldsymbol{y} - \boldsymbol{x})\| \le \frac{L}{2}\|\boldsymbol{x} - \boldsymbol{y}\|^2, \tag{2}$$

$$|f(\boldsymbol{y}) - f(\boldsymbol{x}) - \langle \nabla f(\boldsymbol{x}), \boldsymbol{y} - \boldsymbol{x} \rangle - \frac{1}{2}\langle \nabla^2 f(\boldsymbol{x})(\boldsymbol{y} - \boldsymbol{x}), \boldsymbol{y} - \boldsymbol{x} \rangle| \le \frac{L}{6}\|\boldsymbol{y} - \boldsymbol{x}\|^3. \tag{3}$$

For $\boldsymbol{x}$ and $\boldsymbol{x}^+$ defined as :

$$\boldsymbol{x}^+ \in \arg\min_{\boldsymbol{y}\in\mathbb{R}^d}\left\{ \Omega_{M,\boldsymbol{g},\boldsymbol{H}}(\boldsymbol{y},\boldsymbol{x}) := \langle \boldsymbol{g}, \boldsymbol{y} - \boldsymbol{x} \rangle + \frac{1}{2}\langle \boldsymbol{H}(\boldsymbol{y} - \boldsymbol{x}), \boldsymbol{y} - \boldsymbol{x} \rangle + \frac{M}{6}\|\boldsymbol{y} - \boldsymbol{x}\|^3 \right\}. \tag{4}$$

The optimality condition of equation 4 ensures that :

$$\langle \boldsymbol{g}, \boldsymbol{x}^+ - \boldsymbol{x} \rangle + \langle \boldsymbol{H}(\boldsymbol{x}^+ - \boldsymbol{x}), \boldsymbol{x}^+ - \boldsymbol{x} \rangle + \frac{M}{2}r^3 = 0, \tag{5}$$

where we denoted $r = \|\boldsymbol{x}^+ - \boldsymbol{x}\|$.

It is also known that the solution to equation 4 verifies:

$$\boldsymbol{H} + \frac{M}{2}r\mathbb{I} \succeq 0 \tag{6}$$

We start by proving the following Theorem

**Theorem B.1.** *For any $\boldsymbol{x} \in \mathbb{R}^d$, let $\boldsymbol{x}^+$ be defined by equation 4. Then, for $M \ge L$ we have:*

$$f(\boldsymbol{x}) - f(\boldsymbol{x}^+) \ge \frac{1}{1008\sqrt{M}}\mu_M(\boldsymbol{x}^+) + \frac{M\|\boldsymbol{x}-\boldsymbol{x}^+\|^3}{72} - \frac{4\|\nabla f(\boldsymbol{x})-\boldsymbol{g}\|^{3/2}}{\sqrt{M}} - \frac{73\|\nabla^2 f(\boldsymbol{x})-\boldsymbol{H}\|^3}{M^2}.$$

Using equation 3 with $\boldsymbol{y} = \boldsymbol{x}^+$ and $\boldsymbol{x} = \boldsymbol{x}$ and for $M \geq L$ we have:

$$f(\boldsymbol{x}^+) \overset{\text{equation 3}}{\leq} f(\boldsymbol{x}) + \langle \nabla f(\boldsymbol{x}), \boldsymbol{x}^+ - \boldsymbol{x} \rangle + \tfrac{1}{2}\langle \nabla^2 f(\boldsymbol{x})(\boldsymbol{x}^+ - \boldsymbol{x}), \boldsymbol{x}^+ - \boldsymbol{x} \rangle + \tfrac{L}{6}r^3$$

$$\overset{\text{equation 5+equation 6}}{\leq} f(\boldsymbol{x}) - \tfrac{6M-4L}{24}r^3 + \langle \nabla f(\boldsymbol{x}) - \boldsymbol{g}, \boldsymbol{x}^+ - \boldsymbol{x} \rangle$$

$$+ \tfrac{1}{2}\langle (\nabla^2 f(\boldsymbol{x}) - \boldsymbol{H})(\boldsymbol{x}^+ - \boldsymbol{x}), \boldsymbol{x}^+ - \boldsymbol{x} \rangle$$

$$\overset{M \geq L}{\leq} f(\boldsymbol{x}) - \tfrac{M}{12}r^3 + \langle \nabla f(\boldsymbol{x}) - \boldsymbol{g}, \boldsymbol{x}^+ - \boldsymbol{x} \rangle$$

$$+ \tfrac{1}{2}\langle (\nabla^2 f(\boldsymbol{x}) - \boldsymbol{H})(\boldsymbol{x}^+ - \boldsymbol{x}), \boldsymbol{x}^+ - \boldsymbol{x} \rangle.$$

Using Young's inequality $xy \leq \frac{x^p}{p} + \frac{y^q}{q}$ $\forall x, y \in \mathbb{R}^+$ $\forall p, q > 1$ s.t $\frac{1}{p} + \frac{1}{q} = 1$ we have:

$$\langle \nabla f(\boldsymbol{x}) - \boldsymbol{g}, \boldsymbol{x}^+ - \boldsymbol{x} \rangle \leq \frac{M}{36}r^3 + \frac{2\sqrt{12}}{3\sqrt{M}}\|\nabla f(\boldsymbol{x}) - \boldsymbol{g}\|^{3/2},$$

and

$$\frac{1}{2}\langle (\nabla^2 f(\boldsymbol{x}) - \boldsymbol{H})(\boldsymbol{x}^+ - \boldsymbol{x}), \boldsymbol{x}^+ - \boldsymbol{x} \rangle \leq \frac{M}{36}r^3 + \frac{72}{M^2}\|\nabla^2 f(\boldsymbol{x}) - \boldsymbol{H}\|^3.$$

Mixing all these ingredients, we get

> **Lemma B.2.** *For any $M \geq L$, it holds*
>
> $$f(\boldsymbol{x}) - f(\boldsymbol{x}^+) \geq \frac{M}{36}r^3 - \frac{3}{\sqrt{M}}\|\nabla f(\boldsymbol{x}) - \boldsymbol{g}\|^{3/2} - \frac{72}{M^2}\|\nabla^2 f(\boldsymbol{x}) - \boldsymbol{H}\|^3. \qquad (7)$$

Using equation 2 we have:

$$\|\nabla f(\boldsymbol{x}^+) - \boldsymbol{g} - \boldsymbol{H}(\boldsymbol{x}^+ - \boldsymbol{x}) + \boldsymbol{g} - \nabla f(\boldsymbol{x}) + (\boldsymbol{H} - \nabla^2 f(\boldsymbol{x}))(\boldsymbol{x}^+ - \boldsymbol{x})\| \leq \frac{L}{2}r^2,$$

applying the triangular inequality, we get for $M \geq L$ :

$$\begin{aligned} \|\nabla f(\boldsymbol{x}^+)\| &\leq \tfrac{L}{2}r^2 + \|\boldsymbol{g} + \boldsymbol{H}(\boldsymbol{x}^+ - \boldsymbol{x})\| + \|\nabla f(\boldsymbol{x}) - \boldsymbol{g}\| + \|\nabla^2 f(\boldsymbol{x}) - \boldsymbol{H}\|r \\ &\leq \tfrac{L+2M}{2}r^2 + \|\nabla f(\boldsymbol{x}) - \boldsymbol{g}\| + \tfrac{1}{2M}\|\nabla^2 f(\boldsymbol{x}) - \boldsymbol{H}\|^2 \\ &\leq \tfrac{3M}{2}r^2 + \|\nabla f(\boldsymbol{x}) - \boldsymbol{g}\| + \tfrac{1}{2M}\|\nabla^2 f(\boldsymbol{x}) - \boldsymbol{H}\|^2. \end{aligned}$$

By the convexity of $x \mapsto x^{3/2}$ we have for any $(a_i) \geq 0 : (\sum_i a_i x_i)^{3/2} \leq (\sum_i a_i)^{1/2} \sum_i a_i x_i^{3/2}$, applying this to the above inequality we get

> **Lemma B.3.** *For any $M \geq L$, it holds*
>
> $$\frac{1}{\sqrt{M}}\|\nabla f(\boldsymbol{x}^+)\|^{3/2} \leq 3Mr^3 + \frac{2}{\sqrt{M}}\|\nabla f(\boldsymbol{x}) - \boldsymbol{g}\|^{3/2} + \frac{1}{M^2}\|\nabla^2 f(\boldsymbol{x}) - \boldsymbol{H}\|^3 \qquad (8)$$

We can also bound the smallest eigenvalue of the Hessian. Using the smoothness of the Hessian, we have:

$$\begin{aligned} \nabla^2 f(\boldsymbol{x}^+) &\succeq \nabla^2 f(\boldsymbol{x}) - L\|\boldsymbol{x}^+ - \boldsymbol{x}\|\mathbb{I} \\ &\succeq \boldsymbol{H} + \nabla^2 f(\boldsymbol{x}) - \boldsymbol{H} - Lr\mathbb{I} \\ &\succeq \boldsymbol{H} - \|\nabla^2 f(\boldsymbol{x}) - \boldsymbol{H}\|\mathbb{I} - Lr\mathbb{I} \\ &\overset{\text{equation 6}}{\succeq} -\tfrac{Mr}{2}\mathbb{I} - \|\nabla^2 f(\boldsymbol{x}) - \boldsymbol{H}\|\mathbb{I} - Lr\mathbb{I}. \end{aligned}$$

Which means for $M \geq L$ we have:

$$-\lambda_{min}(\nabla^2 f(\boldsymbol{x}^+)) \leq \tfrac{3Mr}{2} + \|\nabla^2 f(\boldsymbol{x}) - \boldsymbol{H}\|.$$

Then the convexity of $x \mapsto x^3$ leads to the following lemma:

**Lemma B.4.** *For any $M \geq L$, it holds*

$$\frac{-\lambda_{min}(\nabla^2 f(\boldsymbol{x}^+))^3}{M^2} \leq 14Mr^3 + \frac{4}{M^2}\|\nabla^2 f(\boldsymbol{x}) - \boldsymbol{H}\|^3 \tag{9}$$

Now the quantity $\mu_M(\boldsymbol{x}) = \max(\|\nabla f(\boldsymbol{x})\|^{3/2}, \frac{-\lambda_{min}(\nabla^2 f(\boldsymbol{x}))^3}{M^{3/2}})$ which we can be bounded using Lemmas B.3 and B.4:

$$\frac{1}{\sqrt{M}}\mu(\boldsymbol{x}^+) \leq 14Mr^3 + \frac{2}{\sqrt{M}}\|\nabla f(\boldsymbol{x}) - \boldsymbol{g}\|^{3/2} + \frac{4}{M^2}\|\nabla^2 f(\boldsymbol{x}) - \boldsymbol{H}\|^3. \tag{10}$$

Combining Lemma B.2 and equation 10 we get the inequality given in Theorem B.1:

$$f(\boldsymbol{x}) - f(\boldsymbol{x}^+) \geq \frac{1}{1008\sqrt{M}}\mu_M(\boldsymbol{x}^+) + \frac{M}{72}r^3 - \frac{4}{\sqrt{M}}\|\nabla f(\boldsymbol{x}) - \boldsymbol{g}\|^{3/2} - \frac{73}{M^2}\|\nabla^2 f(\boldsymbol{x}) - \boldsymbol{H}\|^3.$$

$$\square$$

### B.2 PROOF OF THEOREM 4.1

To prove Theorem 4.1, we use Theorem B.1 for $\boldsymbol{x} = \boldsymbol{x}_t$, $\boldsymbol{x}^+ = \boldsymbol{x}_{t+1}$, $\boldsymbol{g} = \nabla f(\boldsymbol{x}_t)$ and $\boldsymbol{H}_t = \tilde{\boldsymbol{H}}_t$ the delayed Hessian.

The Hessian error can be decomposed into the error resulting from the delay that we can bound using the second-order smoothness assumption by $L\|\boldsymbol{x}_t - \boldsymbol{x}_{t-\tau_t}\|$, and that arising from the potential inexactness of the Hessian which can be bounded by $\delta_t$ (to account for the initial inexactness); therefore, we obtain the following bound for the iterations of the Algorithm 1 :

$$f(\boldsymbol{x}_t) - f(\boldsymbol{x}^{t+1}) \geq \frac{1}{1008\sqrt{\rho}}\mu_\rho(\boldsymbol{x}^{t+1}) + \frac{\rho}{72}r_t^3 - \frac{73}{\rho^2}\left(\delta_t + L\|\boldsymbol{x}_t - \boldsymbol{x}_{t-\tau_t}\|\right)^3$$

$$\geq \frac{1}{1008\sqrt{\rho}}\mu_\rho(\boldsymbol{x}^{t+1}) + \frac{\rho}{72}r_t^3 - \frac{73 \times 4}{\rho^2}\left(\delta_t^3 + L\|\boldsymbol{x}_t - \boldsymbol{x}_{t-\tau_t}\|^3\right)$$

$$\geq \frac{1}{1008\sqrt{\rho}}\mu_\rho(\boldsymbol{x}^{t+1}) + \frac{\rho}{72}r_t^3 - \frac{73 \times 4}{\rho^2}\delta_t^3 - \frac{73 \times 4}{\rho^2}L^3\left(\sum_{i=t-\tau_t}^{t-1} r_i\right)^3$$

Here we denote $r_t = \|\boldsymbol{x}_{t+1} - \boldsymbol{x}_t\|$.

Rearranging the terms and taking the average over the iteration number $t$, we get the following.

$$\frac{1}{1008T}\sum_{t=0}^{T-1}\mu_\rho(\boldsymbol{x}^{t+1}) \leq \frac{\sqrt{\rho}}{T}\sum_{t=0}^{T-1}\left[f(\boldsymbol{x}_t) - f(\boldsymbol{x}^{t+1}) + \frac{73 \times 4}{\rho^2}\delta_t^3 + \frac{73 \times 4}{\rho^2}L^3\left(\sum_{i=t-\tau_t}^{t-1} r_i\right)^3 - \frac{\rho}{72}r_t^3\right]$$

$$\leq \frac{\sqrt{\rho}}{T}\left[F_0 + \sum_{t=0}^{T-1}\frac{73 \times 4}{\rho^2}\delta_t^3 + \frac{73 \times 4}{\rho^2}L^3\left(\sum_{i=t-\tau_t}^{t-1} r_i\right)^3 - \frac{\rho}{72}r_t^3\right]$$

To deal with the last two terms, we use the following simple technical lemma.

**Lemma B.5.** *For any sequence of positive numbers $\{r_k\}_{k\geq 1}$, it holds for any $m \geq 1$ and $1 \leq \tau \leq m$:*

$$\sum_{k=1}^{m-1}\left(\sum_{i=k-\tau}^{k-1} r_i\right)^3 \leq \frac{\tau^3}{3}\sum_{k=1}^{m-1} r_k^3 \tag{11}$$

*Proof.* We prove equation 11 by induction. It is obviously true for $m = 1$, which is our base.

Assume that it holds for some arbitrary $m \geq 1$. Then

$$\sum_{k=1}^{m}\left(\sum_{i=k-\tau}^{k-1} r_i\right)^3 = \sum_{k=1}^{m-1}\left(\sum_{i=k-\tau}^{k-1} r_i\right)^3 + \left(\sum_{i=m-\tau}^{m-1} r_i\right)^3 \overset{equation\ 11}{\leq} \frac{\tau^3}{3}\sum_{k=1}^{m-1} r_k^3 + \left(\sum_{i=m-\tau}^{m-1} r_i\right)^3.$$
(12)

Applying Jensen's inequality for convex function $t \mapsto t^3$, $t \geq 0$, we have a bound for the second term:

$$\left(\sum_{i=m-\tau}^{m-1} r_i\right)^3 = \tau^3\left(\frac{1}{\tau}\sum_{i=m-\tau}^{m-1} r_i\right)^3 \leq \tau^2\sum_{i=m-\tau}^{m-1} r_i^3.$$
(13)

Therefore,

$$\sum_{k=1}^{m}\left(\sum_{i=1}^{k} r_i\right)^3 \overset{equation\ 12, equation\ 13}{\leq} \left(\frac{\tau^3}{3} + \tau^2\right)\sum_{k=1}^{m} r_k^3 \leq \frac{(\tau+1)^3}{3}\sum_{k=1}^{m} r_k^3. \qquad \square$$

Applying the previous Lemma A.5, we have:

$$\sum_{t=0}^{T-1}\left(\frac{73\times 4}{\rho^2}L^3\left(\sum_{i=t-\tau_t}^{t-1} r_i\right)^3 - \frac{\rho}{72}r_t^3\right) \leq \sum_{t=0}^{T-1}\left(\frac{73\times 4}{3\rho^2}L^3\tau^3 - \frac{\rho}{72}\right)r_t^3$$

We guarantee that this quantity is negative by taking $\rho \geq 20L\tau$.

Thus, for $\rho \geq 20L\tau$, we have:

$$\frac{1}{1008T}\sum_{t=0}^{T-1}\mu_\rho(x^{t+1}) \leq \frac{\sqrt{\rho}}{T}\left[F_0 + \frac{73\times 4}{\rho^2}\sum_{t=0}^{T-1}\delta_t^3\right]$$

$$\leq \frac{\sqrt{\rho}}{T}\left[F_0 + \frac{73\times 4}{\rho^2}\tau_0\delta_0^3 + (T-\tau_0)\frac{73\times 4}{\rho^2}\delta^3\right]$$

Here we have divided the inexactness errors into those coming from the initialization and those coming from the ulterior steps.

All in all, we have:

$$\frac{1}{1008T}\sum_{t=0}^{T-1}\mu_\rho(x^{t+1}) \leq \frac{\sqrt{\rho}F_0}{T} + 73\times 4\frac{\tau_0\delta_0^3}{\rho^{3/2}T} + 73\times 4\frac{(T-\tau_0)\delta^3}{\rho^{3/2}T}. \qquad \square$$
(14)

To obtain the second part of Theorem 4.1, we choose $\rho$ that minimizes the right-hand side of 14 and enforce the conditions $\rho \geq 20L\tau$ and $\rho \geq L$.

To be specific, we choose

$$\rho = \max\left(L, 20\tau L, 18\sqrt{\frac{\tau_0\delta_0^3}{F_0}}, 18\sqrt{\frac{\delta^3 T}{F_0}}\right),$$

For this choice, we get:

$$\frac{1}{1008T}\sum_{t=0}^{T-1}\mu_\rho(x^{t+1}) \leq \frac{\left(1+\sqrt{20\tau}\right)\sqrt{L}\,F_0 + 43\tau_0^{1/4}(F_0\delta_0)^{3/4}}{T} + 43\frac{(T-\tau_0)}{T}\frac{(F_0\delta)^{3/4}}{T^{3/4}}. \qquad \square$$
(15)

## C  HESSIAN INITIALIZATION STRATEGIES

The split-client approach to second-order optimization, particularly in the context of cubic regularization, requires an initial Hessian matrix $H_0$. The choice of $H_0$ contributes to the initial inexactness $\delta_0$ of the curvature estimate. Below are several initialization strategies, ranging from basic to advanced.

## C.1 BASIC INITIALIZATION STRATEGIES

These methods are computationally cheap but provide minimal curvature information.

**Zero Matrix**

- **Strategy:** Set the initial Hessian to the zero matrix.

- **Mathematical Form:**
$$H_0 = \mathbf{0}$$

- **Notes:** This is the simplest choice. The first few steps of the optimization will be largely dictated by the gradient until the first non-zero curvature information is utilized.

**Identity Matrix (Scaled)**

- **Strategy:** Set the initial Hessian to the identity matrix, possibly scaled by a small constant $\mu$.

- **Mathematical Form:**
$$H_0 = \mu \mathbf{I}$$

- **Notes:** This can be viewed as starting the optimization with a fixed-step gradient descent (when $\mu$ is small), providing a stable starting point. It assumes a simple, spherical curvature.

## C.2 ADVANCED INITIALIZATION STRATEGIES

These methods leverage existing knowledge or initial computations to provide a more informed starting point for the curvature.

**Quasi-Newton Approximation (L-BFGS)**

- **Strategy:** Use a few initial gradient steps to construct a **Quasi-Newton** approximation of the Hessian, such as with the **Limited-memory Broyden-Fletcher-Goldfarb-Shanno (L-BFGS)** method.

- **Mathematical Form:** The L-BFGS method maintains a low-rank approximation of the inverse Hessian (or Hessian) based on the history of gradient and parameter updates ($\{s_i, y_i\}$ pairs). $H_0$ is initialized with a simple scaling, typically:

$$H_0^{\text{L-BFGS}} = \frac{y_{k-1}^T y_{k-1}}{y_{k-1}^T s_{k-1}} \mathbf{I}$$

where $s_i = x_{i+1} - x_i$ and $y_i = \nabla f(x_{i+1}) - \nabla f(x_i)$.

- **Notes:** This is a cheap way to get a non-trivial curvature estimate. The split-client framework can benefit from this by initializing $H_0$ with an L-BFGS estimate computed using a small history of initial iterations.

**Problem-Specific Surrogate**

- **Strategy:** Utilize problem-specific knowledge to construct a cheap, yet informative, surrogate for the Hessian.

- **Example (Deep Learning):**

  - **Diagonal Approximation:** Use only the diagonal elements of the true Hessian, which are far cheaper to compute and store (similar to Adagrad/Adam-type methods).
  $$H_0 \approx \text{diag}(\nabla^2 f(x_0))$$

– **Block-Diagonal/Low-Rank Approximation:** For structured models, approximate the Hessian as block-diagonal based on the layer or component structure.

- **Notes:** The quality of $H_0$ is highly dependent on the fidelity of the approximation, but it can offer significant wall-clock speedup if the surrogate is easy to calculate.

**Recycling from a Previous Run**

- **Strategy:** If the current optimization is a restart or part of a sequence of optimizations on similar tasks (e.g., hyperparameter tuning, continuation methods), initialize $H_0$ with the **final Hessian approximation** from the previous, related run.

- **Mathematical Form:**

$$H_0^{\text{new run}} = H_{\text{final}}^{\text{previous run}}$$

- **Notes:** This is the most "informed" initialization, as it leverages full knowledge of the problem's local geometry acquired in a prior computation. This can provide the fastest convergence by minimizing $\delta_0$ from the outset.

## D  CUBIC SUBPROBLEM: PRACTICAL SOLUTION STRATEGIES

Given the current iterate $x_t$, gradient $g_t = \nabla f(x_t)$, a (possibly delayed/inexact) curvature matrix $\widetilde{H}_t$, and cubic parameter $\rho_t > 0$, the step $s_t$ solves the cubic model

$$s_t \in \arg\min_{s \in \mathbb{R}^d} \ m_t(s) \ := \ g_t^\top s + \tfrac{1}{2} s^\top \widetilde{H}_t s + \tfrac{\rho_t}{6} \|s\|^3. \tag{16}$$

The first-order optimality condition is (e.g., (Nesterov & Polyak, 2006; Cartis et al., 2011a))

$$\left( \widetilde{H}_t + \tfrac{\rho_t}{2} \|s_t\| I \right) s_t = -g_t, \quad \text{with} \quad \|s_t\| \geq 0. \tag{17}$$

Pragmatically, one should exploit whatever the curvature client provides (factorizations, matrix–vector products, low-rank structure) to solve equation 16 cheaply and robustly. We summarize practical, implementation-oriented choices below.

**Shifted linear solves + scalar root finding (Moré–Sorensen style).** Define $s(\sigma) := -(\widetilde{H}_t + \sigma I)^{-1} g_t$ for $\sigma \geq 0$ and

$$\phi(\sigma) \ := \ \sigma \ - \ \tfrac{\rho_t}{2} \|s(\sigma)\|. \tag{18}$$

The cubic optimality equation 17 is equivalent to finding the unique root $\sigma^\star \geq 0$ of $\phi(\sigma) = 0$ and then setting $s_t = s(\sigma^\star)$ (Nesterov & Polyak, 2006; Cartis et al., 2011a). The map $\sigma \mapsto \|s(\sigma)\|$ is strictly decreasing, hence $\phi$ is strictly increasing; a safeguarded 1D search (bisection or Newton-bisection hybrid) is therefore simple and reliable:

1. **Bracket.** Obtain a lower bound on the leftmost feasible shift: $\sigma_L \leftarrow \max\{0, -\widehat{\lambda}_{\min}(\widetilde{H}_t)\}$, where $\widehat{\lambda}_{\min}$ is a cheap estimate (e.g., a few Lanczos iterations). Set $\sigma_U \leftarrow \max\{\sigma_L, \sigma_L\}$ and *double* $\sigma_U$ until $\phi(\sigma_U) \geq 0$.

2. **Solve.** Apply bisection (or safeguarded Newton) on $[\sigma_L, \sigma_U]$. Each evaluation of $\phi(\sigma)$ requires solving $(\widetilde{H}_t + \sigma I)u = g_t$ and computing $\|u\|$.

3. **Return.** $s_t \leftarrow -u$ at the final $\sigma$.

*Remark.* The curvature client's **decomposition** can be reused across the 1D search. With an $LDL^\top$ factorization of $\widetilde{H}_t$ (Bunch–Kaufman), solves with $\widetilde{H}_t + \sigma I$ reduce to diagonal shifts of $D$ plus two triangular sweeps; for positive definite cases, a Cholesky of $\widetilde{H}_t + \alpha I$ with a fixed stabilizing shift $\alpha$ is likewise effective.

**Iterative solves when no factorization is available.** If $\widetilde{H}_t$ is accessed via matvecs (e.g., L-BFGS), compute $u(\sigma) \approx (\widetilde{H}_t + \sigma I)^{-1} g_t$ by preconditioned CG or MINRES. Because the 1D search needs several nearby shifts, warm-start each Krylov solve with the previous $u(\sigma)$ and reuse the Krylov basis across shifts (e.g., *recycling* or rational Krylov). Simple preconditioners (diagonal, block-diagonal, or L-BFGS two-loop) are often sufficient. This path keeps the per-step cost near a handful of matvecs, aligning with our claim that once curvature is decomposed/solvable, the cubic step is cheap.

**Handling indefiniteness and negative curvature.** The cubic model is well-defined for indefinite $\widetilde{H}_t$; equation 17 automatically regularizes by the $\frac{\rho_t}{2}\|s_t\|I$ term. In practice:

- Use $LDL^\top$ with symmetric pivoting for robustness on indefinite $\widetilde{H}_t$.

- If a strong negative curvature direction $v$ is detected (e.g., by Lanczos), one can construct a fall-back step along $\text{span}\{-g_t,\ v\}$; a short 1D line search on the cubic model over $s = \alpha d$ (with $d$ in that span) provides a reliable decrease without completing the full root find.

**Warm-starts and reuse across iterations.** Under our split-client setup, the curvature client exposes the same $\widetilde{H}$ across multiple gradient steps until a fresher update arrives. Exploit this by:

1. **Reusing factorizations** of $\widetilde{H}$ (or of $\widetilde{H} + \alpha I$) across many $\sigma$ evaluations and across successive iterations.

2. **Warm-starting** the 1D search at iteration $t$ using the previous $\sigma_{t-1}^\star$; empirically, only a few bracket refinements are needed unless the gradient norm changes sharply.

**Acceptance test and adaptive $\rho_t$.** Although our analysis fixes $\rho_t$ to meet the theoretical conditions, practice benefits from standard trust-region/cubic heuristics (Cartis et al., 2011a; 2012b):

$$r_t := \frac{f(x_t) - f(x_t + s_t)}{m_t(0) - m_t(s_t)}, \qquad \text{accept if } r_t \geq \eta\ (\eta \in (0,1)),$$

and update $\rho_{t+1}$ multiplicatively: increase $\rho$ if $r_t$ is small (model too optimistic), decrease $\rho$ if $r_t$ is large (model conservative). This replaces grid search and improves robustness without sacrificing the asynchronous benefits.

**Stopping criteria and cheap certificates.** In theory, $\mu_{\rho_t}(x_t)$ certifies approximate first/second-order stationarity (Nesterov & Polyak, 2006). In practice, use

$$\|\nabla f(x_t)\| \leq \varepsilon_g \quad \text{and} \quad \widehat{\lambda}_{\min}(\widetilde{H}_t) \geq -\varepsilon_H,$$

where $\widehat{\lambda}_{\min}$ comes from a short Lanczos run (a few matvecs). This aligns with the complexity guarantees while keeping per-iteration overhead negligible.

**Numerical safeguards.** To stabilize solves when $\widetilde{H}_t$ is ill-conditioned or very stale:

- add a tiny ridge $\epsilon I$ to $\widetilde{H}_t$ in the curvature client;

- cap $\sigma$ growth in the 1D search; and

- cap $\|s_t\|$ (e.g., by projecting onto a trust region) if delays are large and the model is potentially outdated.

**Computational cost in context.** With a reusable $LDL^\top$ (or stabilized Cholesky) from the curvature client, each $\phi(\sigma)$ evaluation costs two triangular sweeps plus vector ops; a handful (5–10) of evaluations typically suffices. Hence, *given* curvature decomposition, the cubic step is indeed cheap relative to building/decomposing $\widetilde{H}_t$—matching our time profiles and justifying the delay-centric model that underpins the $\sqrt{\tau}$ wall-clock speedup.

