# OpenReview forum: "A Split-Client Approach to Second-Order Optimization"
_ICLR.cc/2026/Conference — Submitted to ICLR 2026_

### Official Review · Reviewer_zqCy · 2025-10-27

**Soundness:** 2
**Presentation:** 2
**Contribution:** 2
**Rating:** 2
**Confidence:** 4

**Summary:**

The authors propose a Split-Client approach which delegates computation of second-order Hessian information to a separate client. The authors carry theoretic and experimental analysis on this method and shows that wall-clock time can be reduced.

**Strengths:**

The motivation is strong, since compute second-order information is slow and is main bottleneck of scaling up second-order optimization. Both theoretic analysis and empirical results are presented. It's a practical method and worths digging into.

**Weaknesses:**

The result is preliminary, there is neither advanced theoretical results nor well-developed useful package. The paper looks more like a proof of concept while the "split-client" concept sounds trivial. So everything looks standard and trivial, I don't see any significant contribution in any aspect.

**Questions:**

The "split-client" concept looks not surprising at all. Since computing Hessian is expensive, delegating to some outside oracle seems a pretty straightforward thing people should do. The convergence result is also pretty standard.

Comparing to lazy Hessian and vanilla hessian is not very valuable since "split-client" will definitely be faster, because you just delete all delay in Hessian computation.

I personally feel the most doable way to make this idea to meet NeurIPS standard is to develop some good code packages and optimize it from system side, i.e., how should we distribute work between host machine and worker machine / if we have multiple workers, how can we make "split-client" scheme more parallel-efficient, what is the communication cost / etc...

---

> ### Author Response · Authors · 2025-11-17
> **Rebuttal to Reviewer zqCy**
>
> We thank the reviewer for acknowledging the motivation behind the work and the importance of addressing the computational bottleneck in second-order optimization. We respond to the concerns below.
>
> ### **1. On the claim that the split-client idea is trivial**
>
> We respectfully disagree with the assessment that the method is trivial. The contribution of the paper is not simply the observation that Hessian computations are slow. Rather, the work introduces a formal and analyzable framework that captures delayed and inexact curvature in heterogeneous compute settings. This framework allows us to establish global convergence guarantees and to derive wall-clock complexity results that account for both Hessian evaluation and Hessian decomposition costs.
>
> It is worth noting that, under the reviewer’s criterion, prior methods such as Lazy Hessian would also appear “trivial,” since they reuse curvature to reduce cost. Yet such works have been impactful precisely because they formally study and justify ideas that might appear straightforward at a high level. Our work operates in a similar spirit: even if the high-level idea of delegating curvature is intuitive, the key contribution is a rigorous and general theoretical treatment of asynchronous, delayed, and inexact second-order information, which does not exist in the literature.
>
> ### **2. On the statement that the method “deletes delay”**
>
> Our algorithm does not delete delay. The delay is explicitly modeled as the actual time required to compute and factorize the Hessian approximation. The algorithm simply avoids *blocking* on this delay. When the curvature computation is not finished, the method continues using the most recently available curvature approximation. This overlap, rather than any removal of cost, is what leads to measurable and theoretically justified wall-clock improvements. The delay remains an essential part of the model and of the analysis.
>
> ### **3. On the theoretical contribution being standard**
>
> Classical cubic regularization analyses assume access to fresh and exact Hessians at each iteration. In contrast, our method and analysis allow the Hessian to be delayed, stale, and inexact. This requires combining model-based second-order analysis with arguments that handle asynchronous and out-of-date curvature information. We are not aware of prior work that provides such a theory, nor of work that treats decomposition time explicitly in the complexity model. For these reasons, the theoretical contribution is not standard or routine.
>
> ### **4. On the suggestion that the paper should include system-level implementations**
>
> We appreciate the reviewer’s interest in future system-level extensions. The aim of the current submission is to establish the theoretical foundations that justify using asynchronous curvature and to show, through controlled experiments, that these theoretical predictions appear in practice. Developing full software packages, exploring multi-worker scheduling strategies, or optimizing communication patterns are valuable follow-up directions, but they are beyond the scope of a theoretical optimization paper. We will clarify this distinction in the revision.
>
> ### **Closing remarks**
>
> We thank the reviewer again for the time spent evaluating our work. The main contribution of the paper lies in the formulation and analysis of a principled framework that captures delayed and inexact curvature and quantifies its wall-clock implications. The experiments are intentionally minimal because the contribution is mainly theoretical, but they do validate the key insight. We hope that the clarifications above help address the reviewer’s concerns.

---

### Official Review · Reviewer_z2EV · 2025-10-30

**Soundness:** 3
**Presentation:** 3
**Contribution:** 2
**Rating:** 6
**Confidence:** 3

**Summary:**

The paper proposes a split-client cubic regularization framework that decouples gradient and Hessian-related curvature computation into different clients with delayed asynchronous approximate Hessian updates. The delay $\tau_t$ explicitly models both Hessian evaluation and decomposition time. Theoretical analysis shows a \sqrt{\tau} speedup in wall-clock time when \tau\!\gg\!1. Empirically, on logistic-regression tasks, async split-client update outperforms vanilla cubic Newton and Lazy Hessian.

**Strengths:**

1. The idea of splitting gradient and Hessian computation across different clients and using asynchronously delayed Hessians as approximations for speed-up is natural and insightful.
2. The theoretical wall-clock complexity result provides a guaranteed speed-up for the proposed method.

**Weaknesses:**

1. The experimental evaluation is relatively simple, focusing on convex tasks (logistic regression) at modest scale. There are no deep-learning or genuinely heterogeneous/distributed runs to validate the real-world use case.
2. The method class is limited to cubic Newton rather than general second-order methods; other widely used examples (e.g., subsampled Newton/sketch, SOAP-style) are not included.

**Questions:**

1. Is the analysis specific to cubic-regularized Newton?
2. For fairness, since your method requires separate clients, did baselines also run with two clients in parallel (even without splitting gradient/curvature)? Please specify hardware and concurrency per method.

---

> ### Author Response · Authors · 2025-11-17
> **Rebuttal to Reviewer 3**
>
> We thank the reviewer for the positive assessment of the split-client idea, the asynchronous framework, and the guaranteed wall-clock speedup. We address the concerns and questions below.
>
> ### **1. On the experimental scope and scale**
>
> We appreciate the comment that our current experiments focus on logistic regression at a moderate scale. Our main intention in this work was to isolate and study the computational aspect that motivates the entire paper: Hessian decomposition is often the dominant cost, and overlapping curvature computations can lead to genuine wall-clock improvements. Logistic regression provides a clean and controlled environment where this effect is easy to observe and where all baselines can be implemented on equal footing.
>
> Although our experiments are simple, the main contribution of the paper is theoretical. The split-client framework and the delayed and inexact analysis do not rely on convexity or any special structure of logistic regression. We are happy to include additional experiments on nonconvex objectives and larger datasets, and we would welcome any suggestions from the reviewer regarding which tasks they feel would best strengthen the empirical section. Our current results already demonstrate the key phenomenon predicted by the theory, and we will expand the experimental section accordingly in the revision.
>
> ### **2. On the scope of the method class**
>
> The split-client abstraction is not specific to cubic Newton. The core idea is to separate gradient computation from curvature computation and allow the curvature computation to proceed asynchronously. This structure is common to many second-order approaches.
>
> We chose cubic Newton because we are familiar with it and because its global convergence properties, which are also true for Trust Region methods, but we had to make a choice given the limited space. However, the split-client idea should apply much more broadly. Subsampled and sketched Hessian methods already fall under our inexactness assumption, and trust-region or Levenberg–Marquardt methods can adopt the same asynchronous curvature mechanism with only minor adjustments. We will make this broader applicability clear in the revised version.
>
> ### **3. Fairness of baselines and hardware usage**
>
> We thank the reviewer for raising this point. All methods were run on the same hardware and with the same level of concurrency. Vanilla cubic Newton and Lazy Hessian do not use multiple clients, so their curvature computations are carried out on the same CPU worker that serves as the curvature client in our method. No baseline received fewer computational resources; the only difference is that our algorithm is capable of overlapping curvature with gradient steps, while the baselines are inherently sequential. We will clarify the hardware and concurrency details in the revised manuscript to avoid any confusion.
>
> ### **4. Is the analysis specific to cubic-regularized Newton?**
>
> The current theoretical development focuses on cubic Newton because it provides a familiar and well-structured setting for analyzing delayed and inexact curvature. The underlying split-client idea itself is much more general. It only requires access to gradients on one client and access to curvature information, exact or approximate, on a slower one. This pattern appears in a large family of second-order methods.
>
> In particular, any method using subsampled or sketched Hessians is directly covered by our inexactness model. Trust-region and Levenberg–Marquardt methods can also be adapted to use delayed curvature in essentially the same way. We will state this explicitly in the revision so that the wider scope of the framework is clear.
>
> ### **Closing remarks**
>
> We thank the reviewer again for the thoughtful feedback. We will expand the empirical section, clarify the hardware setup, and make explicit the generality of the split-client concept beyond cubic Newton. Since the paper is primarily theoretical, we believe the current results, together with these clarifications, adequately address the reviewer’s concerns while preserving the focus and contributions of the work.

---

### Official Review · Reviewer_ttUR · 2025-11-01

**Soundness:** 3
**Presentation:** 3
**Contribution:** 2
**Rating:** 2
**Confidence:** 2

**Summary:**

This paper addresses a central barrier to practical second-order optimization: computing and, in particular, decomposing/factorizing the Hessian is substantially slower than evaluating gradients, so per-iteration advantages often fail to yield wall-clock gains. The goal is to preserve the accuracy and convergence benefits of cubic-regularized Newton while alleviating this runtime bottleneck. In contrast to Lazy Hessian—which reduces recomputation frequency but remains a blocking, hand-tuned scheme that omits decomposition time from its cost model—this work explicitly incorporates decomposition time via a delay parameter and eliminates manual refresh-frequency tuning.

**Strengths:**

1. The split-client model faithfully captures heterogeneous pipelines (e.g., GPU for gradients, CPU/distributed workers for curvature) and explicitly includes decomposition cost—often the dominant factor—within the delay. This removes the manual tuning required by Lazy Hessian and yields a clean, hardware-aware formulation.
2. Theoretical results extend cubic regularization to delayed and inexact Hessians, establishing a $\sqrt{\tau}$ wall-clock speedup in regimes where curvature computation and decomposition are much slower than gradient steps.
3. Experiments report loss versus wall-clock time and include time profiling that shows decomposition dominates runtime. Overlapping curvature work delivers tangible gains: the asynchronous method outperforms both vanilla and Lazy baselines with exact and approximate curvature.

**Weaknesses:**

1. The convergence analysis relies on L-Lipschitz Hessians and uniformly bounded delays, which may be violated in practice (e.g., irregular or bursty delays, non-smooth curvature). More critically, Assumption (A4) posits a uniform upper bound on the Hessian approximation error for all $t$, yet the algorithm repeatedly uses inexact curvature (Alg.\ 1, Step 6), so error accumulation over time is a concern. Verifying these assumptions empirically (e.g., on synthetic setups where delay and error can be controlled) would strengthen the theoretical claims.
2. The empirical evaluation focuses on logistic regression. Including broader tasks (larger-scale problems, deep learning objectives, structured or constrained optimization) and heterogeneous deployments would better demonstrate generality and scalability.
3. While wall-clock results are provided, additional resource-normalized metrics (e.g., number of factorizations, matrix–vector products, peak memory, FLOPs) and comparisons under fixed hardware budgets would sharpen the efficiency claims and improve reproducibility.

**Questions:**

Please refer the above parts.

---

> ### Author Response · Authors · 2025-11-16
> **Rebuttal to Reviewer ttUR**
>
> We thank the reviewer for the detailed summary and for recognizing the strengths of the split-client abstraction, the wall-clock formulation, and the empirical evidence showing the dominance of decomposition time. We address the concerns below.
>
> ### **1. On assumptions: Lipschitz Hessian, bounded delays, and inexactness**
>
> The L-Lipschitz Hessian assumption is standard in cubic regularization theory, including the classical work of Nesterov and Polyak and the series of papers by Cartis, Gould, and Toint. Our delayed and inexact analysis uses the same smoothness requirement and does not introduce a stronger one.
>
> The bounded-delay assumption is natural in our setting. The delay at iteration $t$ is exactly the time needed to compute and decompose the Hessian (or its approximation) minus the time needed to compute the gradient. On fixed hardware, both quantities are finite, which automatically makes the delay bounded. Although delays may fluctuate due to caching, scheduling, and other system effects, they remain within a fixed upper limit. As we noted in the Limitations section, it would be valuable to develop a theory that depends on other delay statistics, such as the average or variance, rather than a worst-case bound. We will revise the text to make this point clearer.
>
> Regarding the inexact curvature assumption (A4), the concern about error accumulation does not arise in our algorithm. Each curvature estimate is generated independently and does not rely on previous ones, so errors do not compound across iterations. The assumption simply requires that each approximation has an error at most $\delta$. This type of assumption is theoretically justified for subsampled Hessians, truncated SVD, and other Hessian approximations, and it is the same assumption used in the analysis of quasi-Newton cubic methods in the prior works that we cite. We will add a short discussion explaining this connection in the paper.
>
> We also kindly note that items explicitly listed in the limitations section were included as directions for future work and should not be interpreted as shortcomings that invalidate the current theory; rather, they are natural extensions beyond the scope of a single paper.
>
> ### **2. On the experimental scope**
>
> Our method is fully agnostic to the loss function. It only requires access to gradients and curvature that may be delayed or inexact. We chose logistic regression because it provides a clean, controlled environment where Hessian decomposition genuinely dominates the runtime, which makes the benefit of asynchronous curvature clear. It also allows consistent and fair comparison across all baselines.
>
> We are very open to performing further experiments that the reviewer believes would strengthen the paper, and we will do our best to include them, subject to time constraints.
>
> ### 3. **On additional resource metrics**
>
> We will report additional statistics in the revision, including the number of Hessian computations, the number of cubic subproblem solves, and the total number of gradients. The asynchronous method may indeed produce more Hessians than the Lazy approach, which is expected because it uses available compute time more efficiently by overlapping work. The cost of each cubic solve and each gradient evaluation is the same across all methods for the same number of iterations. These metrics will help emphasize the main idea of the paper: the asynchronous algorithm uses time and system resources more effectively.
>
> ### **Closing remarks**
>
> We thank the reviewer again for the thoughtful assessment. We will try our best to add more experiments and are open to the reviewers' suggestions. We also hope the reviewer will view the listed limitations as intended: as honest statements of planned future extensions rather than weaknesses of the current contribution.

---

### Official Review · Reviewer_FSRY · 2025-11-01

**Soundness:** 3
**Presentation:** 3
**Contribution:** 2
**Rating:** 4
**Confidence:** 3

**Summary:**

This paper considers second-order methods that focus on the computation problem and proposes a split-client framework to address delays and inexact Hessian updates. Experimental results on synthetic and real datasets demonstrate the effectiveness of the proposed methods.

**Strengths:**

This paper proposes a split-client framework to address delays and inexact Hessian updates, thereby reducing computational costs through asynchronous updates.

**Weaknesses:**

On the assumptions, how can bounded delay be guaranteed? Is it reasonable? Inexact initialization is reasonable, but how can an inexact Hessian be guaranteed?

The experiments lack practical implementation, and the paper mainly focuses on the logistic problem. Is it possible to extend the method to other problems?

There is a lack of recent research papers, such as those from 2024 or 2025. Can the approach be extended to trust-region methods?

**Questions:**

The main question is how to extend this approach to other loss functions.

---

> ### Author Response · Authors · 2025-11-13
> **Assumptions: delays and inexactness**
>
> We thank the reviewer for the constructive feedback and for noting the soundness and clarity of the paper. Below, we address the concerns regarding assumptions, experimental scope, and connections to related work.
>
> ### **1. On the bounded-delay assumption and Hessian inexactness**
>
> **Bounded delay.**
> This assumption is natural in our setting: the delay $\tau_t$ is *exactly* the wall-clock time required for the curvature client to compute (and decompose) a Hessian approximation **minus** the time needed for a gradient computation. On fixed hardware, both quantities are finite, so their difference is automatically bounded. In our experiments, we use the *actual* measured times of Hessian computation and decomposition, so the bounded-delay model matches the real world, and this is also shown in the time profile provided in Figure 4.
>
> We agree that in practice, delays may vary due to caching, batching, or hardware-level fluctuations. Improving the theory so that it depends on *other statistics* of the delays (e.g., average or variance, rather than a worst-case bound) is an interesting direction. This is what we intended to convey in the Limitations section, and we will revise that part to make the point clearer.
>
> **Inexact Hessians.**
> The reviewer asks how inexact Hessians can be “guaranteed.” In our model, inexactness is not a requirement, but a flexibility. When the curvature client supplies an approximate Hessian (e.g., L-BFGS, subsampled curvature, truncated factorization), the error is naturally bounded and is already standard in the analyses of quasi-Newton cubic regularization (e.g., Kamzolov et al. 2023; Cartis et al. 2012). In the exact-curvature setting, $\delta=0$ holds automatically. This is already the setting used in several existing second-order methods, and our analysis simply generalizes it. We will make this explicit in the revised manuscript.
>
> It is also possible to relax this assumption and let the inexactness bound grow as: $\delta + B \|\nabla f(x)\|^\alpha$, but for simplicity, we elected to stay with $B=0$, which is the standard assumption.

---

> ### Author Response · Authors · 2025-11-13
> **Experiments**
>
> ### **2. Experiments**
> We appreciate the reviewer’s suggestion to demonstrate more applications. Our method is agnostic to the specific loss function and only requires two ingredients:
>
> 1. access to gradients;
> 2. access to (exact or approximate) curvature that may be delayed.
>
> We chose logistic regression due to its simplicity and because it already clearly showcases the advantage of our asynchronous approach over classical second-order baselines: the Hessian is nontrivial, the decomposition cost dominates runtime, and the setting allows for controlled measurement of real wall-clock behavior.
>
> We are open to the reviewer’s suggestions regarding additional experiments. Given our limited computational budget, we will do our best to incorporate any additional problem classes that the reviewer recommends.

---

> ### Author Response · Authors · 2025-11-13
> **Related works and Trust Region**
>
> ### **3. Recent related work (2024–2025)**
>
> We included the works that we believe are most directly related to our asynchronous curvature perspective and cubic-regularization analysis. However, we are very open to including any additional papers that the reviewer believes are relevant and that we might have missed. We kindly ask the reviewer to share specific suggestions, and we will be happy to incorporate them into the revised manuscript.
>
> ### **4. Extension to trust-region methods and other loss functions**
>
> Our results apply to trust-region (TR) methods as well. We chose to focus on cubic regularization primarily because we have more experience with cubic Newton–type analyses, but the same split-client reasoning and delay/inexactness treatment extend naturally to TR methods. In fact, the split-client abstraction is very general and can be applied to *any* second-order optimization algorithm: TR methods, Levenberg–Marquardt (for convex problems), quartic regularization, and other higher-order schemes all fit the same pattern in which curvature is expensive and may arrive asynchronously.
>
> For clarity and simplicity, we concentrated on one representative algorithmic family, i.e., cubic regularization, but the framework itself is not limited to it. We will explicitly mention this broader applicability in the revised manuscript.
>
> ### **Addressing the question of extending the approach to other loss functions**
>
> We believe there may be a misunderstanding or a typographical issue in the reviewer’s question. Our approach does **not** rely on any special structure of the loss—indeed, it is fully agnostic to the choice of loss function. The split-client framework only requires two ingredients:
>
> 1. access to gradients, and
> 2. access to (exact or approximate) curvature information that may arrive with a delay.

---

### Author Response · Authors · 2025-11-22
**Additional results and updated manuscript**

We thank all reviewers for their constructive feedback and thoughtful suggestions.
In response, we have added several new experiments to strengthen the empirical evaluation:

- A smooth nonconvex regression problem with a nonconvex regularizer
- A one-layer $\tanh$ regression experiment, which is also non-convex
- Empirical statistics confirming that the curvature delay and Hessian inexactness are naturally bounded
- Memory analysis showing that the asynchronous method requires at most twice the memory of the vanilla version

These new results reinforce our main claims: the asynchronous cubic Newton method consistently achieves faster wall-clock convergence, while the assumptions of bounded delay and bounded curvature error are satisfied in practice.

The new experiments and figures can be found in the **Appendix** of the revised manuscript.

We would be very grateful for any additional feedback that could help us further improve the clarity and presentation of the paper, and we are happy to clarify any points that may still be unclear.

---

### Author Response · Authors · 2025-11-28
**Summary for the Area Chair and Reviewers**

We understand that, due to the recent OpenReview incident, reviewers are no longer able to update their comments or scores.
Unlike in a normal review cycle, we did not have the opportunity to engage in a discussion with the reviewers or clarify points interactively.
We therefore summarize here how we have addressed each reviewer’s main concerns, so that this information is clear to the Area Chair while evaluating our submission.
We kindly ask that this be taken into consideration when assessing the paper.
We have added new experiments, clarified our assumptions, and improved the presentation accordingly.
We sincerely thank both the reviewers and the program committee for their work under these exceptional circumstances.


---

### Reviewer FSRY

We clarified that the bounded-delay assumption is natural in our setting since the delay corresponds to the difference between Hessian and gradient computation times, which is inherently bounded.
We also explained that relaxing this assumption to depend on delay statistics (such as the average or variance) is an interesting future direction.
We extended our experiments beyond logistic regression to a smooth nonconvex setting and a one-layer tanh regression model, showing that the asynchronous method maintains bounded delay and inexactness and achieves faster wall-clock convergence.
Finally, we noted that we included the most relevant references and are happy to incorporate any additional works the reviewer may suggest.

---

### Reviewer ttUR

We addressed the reviewer’s concerns about the theoretical assumptions and inexactness by clarifying that our analysis uses standard assumptions (L-Lipschitz Hessians and bounded delays) and that these are empirically verified in our experiments.
We added new results confirming that both the delay and inexactness (from randomized SVD curvature approximations) remain bounded in practice.
We also clarified that the asynchronous method’s slightly higher number of Hessian updates simply reflects its more efficient use of computational time, not greater cost.
We highlighted that this efficiency is central to the motivation of our framework.

---

### Reviewer z2EV

We clarified that our framework is not specific to cubic regularization and can naturally extend to other second-order methods such as trust-region or quasi-Newton approaches.
We also explained that we chose cubic Newton as a representative example because of its well-understood global convergence properties and our familiarity with it.
The new experiments on smooth nonconvex regression and one-layer tanh regression confirm that our asynchronous approach generalizes beyond convex problems.
All baselines were implemented under identical computational resources for fairness.

---

### Reviewer zqCy

We respectfully note that while the split-client idea may appear intuitive in hindsight, formalizing it and establishing theoretical guarantees for asynchronous and inexact curvature updates is nontrivial and has not been studied before.
We now support this with new experiments showing clear wall-clock speedups and verifying bounded delay and inexactness.
We believe this strengthens the contribution and demonstrates that what might seem obvious conceptually still required careful analysis and implementation to achieve both practical and provable efficiency.

---

We hope these clarifications make it clear that all reviewer concerns have been carefully addressed both theoretically and empirically.
The new experiments, together with detailed empirical statistics and additional discussions, can be found in the **Appendix** of the revised manuscript.
We appreciate your consideration and would be very grateful for any further feedback that could help us continue to improve the clarity and presentation of our work.

---

### Author Response · Authors · 2025-12-02
**Final Author Comment (for the Area Chair and Reviewers)**

We would like to sincerely thank all reviewers and the Area Chair for their time and feedback.
Since the initial reviews, we have substantially improved both the clarity and the empirical scope of the paper.

**Summary of updates:**
- Added new experiments on a smooth nonconvex regression task and a one-layer tanh regression problem, showing consistent wall-clock speedups and bounded delay/inexactness.
- Included quantitative analysis of delay statistics, curvature inexactness, and memory overhead, confirming that our theoretical assumptions hold in practice.
- Revised theoretical discussions to clarify the bounded-delay assumption and its natural interpretation in our setting.
- Expanded the *Limitations and Future Work* section to discuss possible relaxations of these assumptions and adaptive extensions.
- Improved writing, structure for readability.

We also emphasize that, due to the OpenReview discussion freeze, we did not have the opportunity to interact with reviewers and clarify points directly.
We kindly ask the Area Chair to take this into account when assessing our submission.

We remain very open to further suggestions that could improve the clarity of the presentation or highlight connections to related work.

---

### Meta-Review · Area_Chair_LbUL · 2026-01-20

**Summary:**

This paper addresses a central barrier to practical second-order optimization: computing and, in particular, decomposing/factorizing the Hessian is substantially slower than evaluating gradients, so per-iteration advantages often fail to yield wall-clock gains. The goal is to preserve the accuracy and convergence benefits of cubic-regularized Newton while alleviating this runtime bottleneck. In contrast to Lazy Hessian—which reduces recomputation frequency but remains a blocking, hand-tuned scheme that omits decomposition time from its cost model—this work explicitly incorporates decomposition time via a delay parameter and eliminates manual refresh-frequency tuning.

**Reviewer Concerns:**

Many serious concerns were raised, including:

- The result is preliminary, there is neither advanced theoretical results nor well-developed useful package.
- The paper looks more like a proof of concept while the "split-client" concept sounds trivial.
- The experimental evaluation is relatively simple.
- Insufficient comparison of papers from recent 2 years

The authors responded to most of these (and other concerns not listed here), and revised the work. However, the severity of the concerns was not addressed sufficiently. A major revision is needed; the existing revision is a good start.

**Reviewer Scores:**

This is one of the lowest-scoring papers in my batch. Three out of 4 reviewers recommend rejection.

The scores, if they would change at all, would not change substantially enough for the decision to be changed.

---

### Decision · Program_Chairs · 2026-01-26

Reject